# Patient-Reported Outcomes of Endovascular Treatment of Post-Thrombotic Syndrome: Ancillary Study of a French Cohort

**DOI:** 10.3390/diagnostics13142357

**Published:** 2023-07-13

**Authors:** Kévin Guillen, Frédéric Thony, Costantino Del Giudice, Gilles Goyault, Arthur David, Frédéric Douane, Yann Le Bras, Valérie Monnin-Bares, Jean-François Heautot, Hervé Rousseau, Thomas Martinelli, Francine Thouveny, Pierre-Antoine Barral, Vincent Le Pennec, Pascal Chabrot, André Rogopoulos, Ludwig Serge Aho-Glélé, Marc Sapoval, Mathieu Rodière, Olivier Chevallier, Nicolas Falvo, Romaric Loffroy

**Affiliations:** 1Department of Interventional Radiology, CHU Dijon, 21000 Dijon, France; kevin.guillen@chu-dijon.fr (K.G.);; 2Department of Interventional Radiology, CHU Grenoble, 38000 Grenoble, France; 3Interventional Radiology, Institut Mutualiste Montsouris, 42 Boulevard Jourdan, 75014 Paris, France; 4Department of Vascular and Oncological Interventional Radiology, Institut Cardiovasculaire de Strasbourg (ICS), Clinique Rhena, 67000 Strasbourg, France; 5Department of Interventional Radiology, CHU Nantes, 44000 Nantes, France; 6Department of Radiology, Pellegrin Hospital, Place Amélie Raba Léon, 33076 Bordeaux, France; 7Department of Imaging and Interventional Radiology, Montpellier University Hospital (CHU), 34000 Montpellier, France; 8Radiology Department, University Hospital Pontchaillou, 35000 Rennes, France; 9Cardiac Imaging Centre, Toulouse University Hospital, 31000 Toulouse, France; 10Department of Medical Imaging and Radiology, Valence Hospital, 179 bd Maréchal Juin, 26953 Valence, France; 11Vascular Radiology, University Hospital, 49000 Angers, France; 12Department of Radiology, La Timone Hospital, Assistance Publique des Hôpitaux de Marseille, 13000 Marseille, France; 13Department of Interventional and Diagnostic Imaging, University Hospital of Caen, Avenue de la Côte de Nacre, 14033 Caen, France; 14Department of Vascular Radiology, Hôpital Gabriel Montpied, CHU Clermont-Ferrand, Place Henri Dunant, 63000 Clermont-Ferrand, France; 15Department of Radiology, Institut Arnault Tzanck, 06700 Saint-Laurent du Var, France; 16Department of Epidemiology, Statistics and Clinical Research, Hôpital Universitaire François-Mitterrand, 21079 Dijon, France; 17Vascular and Oncological Interventional Radiology Department, Assistance Publique-Hôpitaux de Paris, Hôpital Européen Georges-Pompidou, 75015 Paris, France

**Keywords:** post-thrombotic syndrome, endovascular treatment, angioplasty, stenting, patient-reported outcomes

## Abstract

Excellent outcomes of angioplasty/stenting for the post-thrombotic syndrome (PTS) have been reported, notably regarding objective criteria in the vast French SFICV cohort. Differences may exist between patient-reported and objective outcomes. We investigated this possibility by using validated scales because significative correlations are discordant in the literature between patency and patient-reported characteristics. Patient-reported outcomes seem to be a more consistent tool than radiologic patency for the diagnosis and follow-up of patients displaying PTS. We retrospectively reviewed the Villalta scale and 20-item ChronIc Venous dIsease quality-of-life Questionnaire (CIVIQ-20) scores recorded after endovascular stenting for PTS at 14 centres in France in 2009–2019. We also collected patency rates, pre-operative post-thrombotic lesion severity, and the extent of stenting. We performed multivariate analyses to identify factors independently associated with improvements in each of the two scores. The 539 patients, including 324 women and 235 men, had a mean age of 44.7 years. The mean Villalta scale improvement was 7.0 ± 4.7 (*p* < 0.0001) and correlated with the thrombosis sequelae grade and time from thrombosis to stenting. The CIVIQ-20 score was available for 298 patients; the mean improvement was 19.2 ± 14.8 (*p* < 0.0001) and correlated with bilateral stenting, single thrombosis recurrence, and single stented segment. The objective gains demonstrated in earlier work after stenting were accompanied by patient-reported improvements. The factors associated with these improvements differed between the Villalta scale and the CIVIQ-20 score. These results proved that clinical follow-up with validated scores is gainful in patients treated for PTS thanks to a mini-invasive procedure.

## 1. Introduction

Chronic venous insufficiency affects 1–5% of adults and may be primary or a consequence of deep vein thrombosis (DVT), usually at a lower limb [1]. Despite conservative treatment with anticoagulants and compression stockings or bandages, post-thrombotic syndrome (PTS) develops in 23% to 60% of patients within two years after DVT [2,3]. The main symptoms of PTS are chronic pain and oedema of the lower limb. Severe forms, which occur chiefly after ilio-femoral DVT, account for only 5% to 10% of cases but impair the quality of life and place an economic burden on society [3,4,5,6,7,8]. In PTS, chronic venous hypertension is due to venous obstruction or stenosis combined with valvular incompetence. This causes venous reflux and venous outflow obstruction [6]. Abnormalities of the microcirculation and lymphatic system also develop. They include valvular dysfunction and micro-valvular destruction in venules with reflux, vein wall lesions as synechiae, and overloading of the lymphatic system. All these lesions can lead to the congestion of drained tissues, with patients displaying symptoms such as pain, cramps, and pruritus or clinical signs, such as oedema or skin induration [7,8].

The treatment seeks to remove the obstruction to venous outflow. Endovascular techniques emerged in the mid-1990s, and their uses are now supported by a high level of evidence [9]. The few contra-indications to ilio-caval stenting are non-correctable coagulopathy and local or systemic infection [1]. Endovascular methods are less invasive than surgery and require lower levels of post-therapeutic care, including a faster discharge. Two kinds of complications that are extremely rare are described. One of these is venous rupture, which can be treated in the same procedural time with covered stents. On the other hand, thrombotic complications can occur, including stent thrombosis and pulmonary embolism, which can be prevented by anticoagulant therapy that is systematically considered during the procedure [3]. The effectiveness is typically assessed using objective criteria, such as patency by ultrasonography (US) or computed tomography (CT) [10]. The French Society of Cardio-Vascular Imaging (SFICV) has reported excellent objective short and mid-term outcomes of stenting for PTS in a vast observational cohort. They studied stent patency as the primary outcome with a mean follow-up of 21.0 months. They showed that primary patency, primary assisted patency, and secondary patency were achieved in 80.4%, 84.7%, and 92.2% of the 668 included patients, respectively [3]. 

Patient-reported outcomes are also important to consider [11] and do not always mirror objective treatment results. The Villalta score recommended by the American Heart Association for assessing PTS severity includes subjective symptoms and clinical signs that can be assessed by a physician. They include congestive signs [12,13,14]. A specific quality-of-life assessment tool for chronic venous insufficiency, the 20-item ChronIc Venous dIsease quality-of-life Questionnaire (CIVIQ-20), was developed in France and validated internationally for PTS assessment [15,16,17].

Here, we conducted an ancillary analysis of the data from the SFICV cohort to assess patient-reported outcomes using the Villalta and CIVIQ-20 scores. 

## 2. Materials and Methods

### 2.1. Study Design

We conducted an ancillary study of the data from the retrospective multicentre SFICV cohort study of patients with PTS managed by endovascular stenting under real-life conditions [3]. The 15 participating centres were university, community, or private hospitals. The protocol for the initial study [3] was approved by the French data protection agency and the appropriate ethics committee (CERF-CERIM and IRB CRM-1911-057, respectively). In accordance with French regulations about retrospective non-interventional studies of anonymised data, informed consent was not required for the present study. Informed consent was obtained from each patient before the endovascular procedures. 

### 2.2. Study Population

The initial study included patients who had PTS due to chronic post-thrombotic proximal lower limb venous occlusion at least three months after acute caval or ilio-femoral DVT. We selected patients displaying disabling symptoms with inadequate effectiveness of anticoagulant therapy and optimal compression [3]. Endovascular stenting was performed between January 2009 and December 2019. Figure 1 illustrates the diagnosis and treatment procedure. The exclusion criteria were acute thrombosis, extrinsic compression by a tumour, non-thrombotic obstruction, Budd-Chiari syndrome, and venous occlusion at the site of a dialysis catheter because the physiopathologic mechanisms are not strictly the same in these conditions.

Of the 698 patients in the initial cohort, 668 had a technically successful procedure, defined as successful recanalization, and stent deployment, restoring rapid anterograde flow into the targeted vessel, with stent patency confirmed by Doppler US before discharge [10]. The procedure was classically performed under sedation, as described previously, through a venous approach depending on the interventionalist’s habitus, including popliteal, jugular, or femoral vein puncture [3]. Among these 668 patients, 539 and 298 had available Villalta and CIVIQ-20 scores, respectively, which were used for the current study.

After stenting, treatment with local compression and antithrombotic agents was at the discretion of each managing physician.

### 2.3. Data Collection

For the initial study, the medical records were reviewed retrospectively, and the information for each patient was entered into a database. The age, sex, mean time from first acute DVT to endovascular treatment, thrombosis recurrence, thrombophilia, any underlying risk factors, and development of post-thrombotic venous abnormalities were assessed by colour duplex US or CT venography with the CT-lesion severity grade [18]. The extension and location of the PTS venous were recorded. The CT severity grade was established according to the scale proposed by Menez et al. based on the vein diameter, synechiae, and possibility of stenting, with four stages of severity from grade 0 (no lesion) to grade 3 (major lesions) [18]. The endovascular procedure characteristics, including the number of stents and the stented venous segments identified according to the lower extremity thrombosis classification (LET), were also assessed [19]. 

### 2.4. Villalta and CIVIQ-20 Scores

The Villalta scale and CIVIQ-20 are extensively validated tools [2,15,16,17,18,20]. The Villalta scale was used to evaluate the PTS severity at baseline and during the follow-up. The scale items consist of five subjective symptoms (pain, heaviness, cramps, pruritus, and paraesthesia) and six physical findings (oedema, induration, hyperpigmentation, new venous ectasia, redness, and pain on calf compression), each rated from 0 (absent) to 3 (severe). In patients with bilateral lesions, the most disabling limb was scored. A total score of 0 to 4 indicates the absence of PTS, 5 to 9 indicates mild PTS, 10 to 14 indicates moderate PTS, and ≥15 indicates severe PTS [17]. The patients completed the CIVIQ-20 self-questionnaire at baseline and during the follow-up [16]. The 20 items assess the perceived impact of the venous disease on four quality-of-life dimensions: pain (four items), physical well-being (four items), psychological well-being (nine items), and social life (three items). Each item is scored from 1 to 5, with higher scores indicating greater quality-of-life impairments. This questionnaire was developed by Professor Launois with an educational grant from SERVIER^®^ [15,16].

The Villalta and CIVIQ-20 scores were evaluated immediately before intervention and during the follow-up (at 1, 6, 12, 36, and 60 months) to assess the clinical severity of PTS. Only the last available score data were considered for the analyses.

### 2.5. Statistical Analyses

All the statistical analyses were performed using STATA software, version 15.1 (STATA Corp., College Station, TX, USA). The qualitative variables were described as proportions and percentages. The semi-quantitative variables were described as the median [interquartile range] and the continuous variables were described as the mean ± SD (range). The paired *t* test was used for univariate analysis of the Villalta and CIVIQ-20 scores, which were evaluated separately. The Villalta score was available for 539 patients and the CIVIQ-20 score was available for 298 patients. Although both scores are semi-quantitative variables, they were handled as continuous variables given the large sample size and number of modalities after validating the convergence of parametric and non-parametric approaches. Only the results of the parametric tests are reported. Values of *p* < 0.05 were considered significant.

Two multivariate linear regression models were built, one for the Villalta score and the other for the CIVIQ-20 score, using manual backward stepwise selection. Various intermediate models were constructed. Only the final optimal model is reported. This procedure ensured the robustness of our analysis despite the exclusion of potentially relevant variables, such as the centre, sex, and age. 

## 3. Results

### 3.1. Characteristics of the Patients and Procedures

The 539 patients with Villalta score data were recruited at 14 centres. Table 1 shows their main features. All 298 patients with CIVIQ-20 score data also had Villalta score data. The median time from stenting to the last imaging study was 16.1 [8.6–32.2] months. Table 2 provides information on the endovascular procedure. Other data can be particularly interesting for comparison to published papers, as the left/right symptomatic limb ratio was calculated as 4.8/1 for 82 (15%) patients that had an associated contralateral lesion (bilateral obstruction). The median age of the patients was 43 years (IQR 32.3–56.5) with a sex ratio of 1.5/1. CT grading sequelae were realized for all the presented patients. In total, 151 (28%) patients had one or more thrombosis recurrences.

### 3.2. Changes in the Villalta and CIVIQ-20 Scores

The Villalta score improved significantly from the baseline to the post-procedural assessment. The mean score improvement by univariate analysis was 7.0 ± 4.7 (*p* < 0.00001) (Figure 2). As shown in Table 2, only 25 patients had severe PTS after stenting. Of the 18 patients in the no PTS Villalta score category before stenting, none were in the PTS category after stenting.

The CIVIQ-20 score improved significantly, as shown in Table 1 and Table 2 (*p* < 0.0001) (Figure 3). Of the 18 patients in the no PTS Villalta score category before stenting, 14 had improved CIVIQ-20 scores after stenting.

### 3.3. Factors Associated with Changes in the Villalta and CIVIQ-20 Scores

By multivariate analysis, the factors independently associated with a Villalta score improvement after stenting were grade 3 DVT sequelae; six or seven DVT recurrences; a shorter median time from DVT to stenting; and stenting of six or seven segments (Table 3). The factors independently associated with a CIVIQ-20 score improvement were bilateral initial thrombosis, stenting of a single segment, and a single DVT recurrence (Table 4).

## 4. Discussion

Our study showed that stenting to treat PTS was followed by statistically significant improvements in the Villalta score, which includes five subjective symptoms, and the quality of life as assessed using the CIVIQ-20 score. Sarici et al. and Ye et al. reported a significant decrease in the Villalta score, respectively, from 18 and 22 at baseline to 8 and 9 after the procedure in accordance with our findings [14,20,21]. A quality-of-life improvement was only observed for the work by Sarici et al., which displayed a significant improvement in the patient’s feelings and was in agreement with our data [13]. 

As demonstrated in the literature, combining two clinical scores for this purpose is valuable. Both the Villalta scale and the CIVIQ-20 score have been validated internationally for symptoms and signs evaluation and for Quality-of-Life (QOL) assessment in PTS. The Villalta scale allows the gradation of PTS severity considering clinical characteristics, whereas the CIVIQ-20 documents daily life alteration linked to PTS. In keeping with our findings, previous studies documented Villalta score improvements after stenting with similar independent associations to those in our population [2,18,22]. It is worth noting, however, that some patients have disabling symptoms not included in the Villalta score [2,13,23,24]. Of our 539 patients, 18 were not categorised by the Villalta score as having PTS and improved their CIVIQ-20 score. Our results confirmed the complementarity of the two scores, particularly in borderline patients, including those with a mild Villalta scale displaying disabling symptoms unexpectedly [2,22,23]. The Villalta scale for the PTS severity grade and the CIVIQ-20 score for the quality of life are mandatory in PTS. The multivariate analyses of the patient-reported scores did not entirely corroborate the findings of David et al. Other factors were independently associated with improvements. They differed markedly between the Villalta and CIVIQ-20 scores, and patients’ feelings were more associated with anamnesis variables than imaging data [1,9,21,24,25,26,27,28]. We demonstrated that there is an effect of the sequelae grade (significant result only for grade 3), a high number of thrombosis recurrences, and confirmed that the median delay from DVT and stenting extension, particularly from six to more segments (including bilateral lesions de facto), was independently associated with Villalta scale improvement [2,18,22]. A study of 52 patients also used the CIVIQ-20 score and showed significant gains [13]. A CIVIQ-20 score improvement was independently associated with bilateral initial thrombosis, stenting of a single segment, and a single thrombosis recurrence. These results are in contrast with those regarding the Villalta score, which more often improved in patients with many DVT recurrences and stenting of multiple segments. The two tools do not measure the same items, and only the CIVIQ-20 is specifically designed to assess the quality of life. Moreover, only 298 of the 539 patients with Villalta score data were recorded with CIVIQ-20 score data, which could also explain the differences. A past history of recurrent thrombosis and the delay from DVT were independent factors of clinical score improvement, whereas the stenting patency assessed by imaging was not. This proved the limit of CT or US scan outcomes if considered alone [2,3,9]. In contrast to others, we demonstrated no effect of age on the score improvement [5,26]. 

Several studies have demonstrated the effectiveness of endovascular stenting for PTS [22,24]. This procedure is very safe, with no reports, to our knowledge, of peri-operative mortality or pulmonary embolism [11,14,22,25]. Moreover, except for medical reasons, the first approach of these procedures is now performed under local anaesthesia as a day-hospital procedure [29]. Selection criteria must be applied to limit risk and increase the potential efficiency of the procedure. Focused narrowing at the ilio-femoral level with few synechiae proximally at the femoral level is more likely to result in a satisfying stent inflow that is correlated with the clinical success of the procedure. This is well-screened by the sequelae grade classification developed by Menez et al. [18]. In some patients, an endovascular approach cannot be performed, and surgery is mandatory to treat PTS. The selection of these patients should be optimal, particularly considering anaesthetic and surgical risks [3,10,30,31,32,33,34,35]. 

After stenting, patients are usually given anticoagulation for two to six months associated with aspirin for one month. Long-term therapeutic anticoagulation is typically given to patients with known hypercoagulability, whereas no specific antithrombotic drug is required for others [11,32]. Many factors, such as venous inflow, stent characteristics, or stent localization, may affect venous patency and, thus, the risk of adverse events. The role of periprocedural antithrombotic management in the preservation of stent patency is not clearly defined [18]. Due to the heterogeneity of the treatment strategy and its connection to past medical history, only systematic reviews have been published. To our knowledge, there are no Class I European Guidelines specifying the duration and type of antithrombotic drug (antiplatelet and/or anticoagulant) in the case of venous stenting even if clinical benefits can be presupposed [33]. Note that studies in the cardiac field prove that for patients with coronary heart disease, when combined with antiplatelets, the safety and efficacy of different anticoagulants are controversial. Indeed, the risk of bleeding must be considered for long-term treatment. The venous regimen is different from the arterial one, with slower pressure and blood flow that can potentialize the hypercoagulability state linked to recent intervention with stent placement, particularly in patients with thrombophilia [34]. Based on empirical data and local habitus, most physicians give therapeutic anticoagulation for 2 to 6 months, followed by aspirin for a non-determined time. Long-term therapeutic anticoagulation is typically given to patients with a hypercoagulable state requiring lifelong anticoagulation [12,32,35]. Moreover, some authors highlight the benefits of collaborative work only in complex situations, while others conclude that a daily collaborative network including angiologists and interventional practitioners seems to be the better way to improve the benefits of the patient from the procedure [34,35]. Nowadays, special attention should be given to associated measures that are mandatory to ensure good results in the long term. For example, authors have described the intermittent use of pneumatic compression stockings until a patient is completely mobilized after successful recanalization [36]. Otherwise, the antithrombotic medication from acute DVT must be carefully chosen to avoid PTS and intrastent restenosis or bleeding complications. Indeed, accumulated evidence demonstrated that the treatment of acute DVT with DOACs (Direct Oral Anticoagulants) rather than warfarin was associated with a numerically lower but statistically non-significant risk of PTS.

The grade sequelae classification proposed by Menez et al. based on therapeutic objectives appeared relevant in our mind. It extrapolated stenting inflow as a prognostic factor to preserve good patency and, thus, a great clinical result [18]. Moreover, the choice of stent type depends in part on the location of the segment or segments to be stented. Local venous mapping is, therefore, crucial. Even if the stenting data were not listed systematically in this retrospective cohort, the past decade has seen approvals of new devices and a toolbox that keeps evolving. The localization of the stenting appeared as a key point to choose the fitting device. For example, Venovo™ and Sinus Obliquus™ stents for symptomatic ilio-femoral venous disease displayed a significant patency rate and clinical improvement with a low complication rate [37]. The Blueflow Venous Stent is more dedicated to be placed across the inguinal ligament [38]. Intravascular US during the procedure may also help to achieve successful stenting, allowing a better choice of landing areas and screening for earlier potential side effects [12,39].

The limitations of our study include the retrospective design. Thus, CIVIQ-20 data were missing for 241 of the 539 patients whose Villalta scales were available. Superficial venous disease was not consistently recorded but can affect the clinical scores. The body mass index was not collected. The choice of stent type and antithrombotic treatment strategy was not standardised because no high-grade recommendations exist. The scores were determined at varying time points depending on the delay from the procedure to the study endpoint.

## 5. Conclusions

Symptoms of PTS and the quality of life improved significantly after stenting. These results proved that combining clinical follow-up with validated scores can be useful in patients treated for PTS thanks to a mini-invasive procedure, even if imaging has to be considered to document the underlying vein characteristics’ evolution with time. Prospective studies with standardised approaches of stent selection and antithrombotic medications are warranted to further assess the outcomes of stenting used to treat PTS. 

## Figures and Tables

**Figure 1 diagnostics-13-02357-f001:**
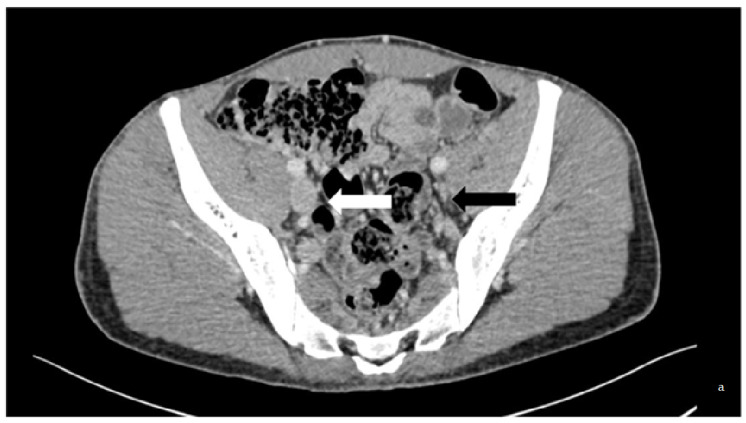
Case of a 33-year-old patient with thrombophilic disease whose first DVT was diagnosed in 2013. He displayed PTS. (**a**) is a transversal slice of a CT scanner that shows the baseline phleboscanner with a collapsed left external iliac vein (black arrow) compared to the collateral one (white arrow) confirmed on (**b**), which shows phlebography assessment in a posterior incidence (black arrow). The treatment consisted of an angioplasty/stenting of the iliofemoral vein by popliteal vein approach, the results of which are displayed on the digital subtraction angiography (**c**).

**Figure 2 diagnostics-13-02357-f002:**
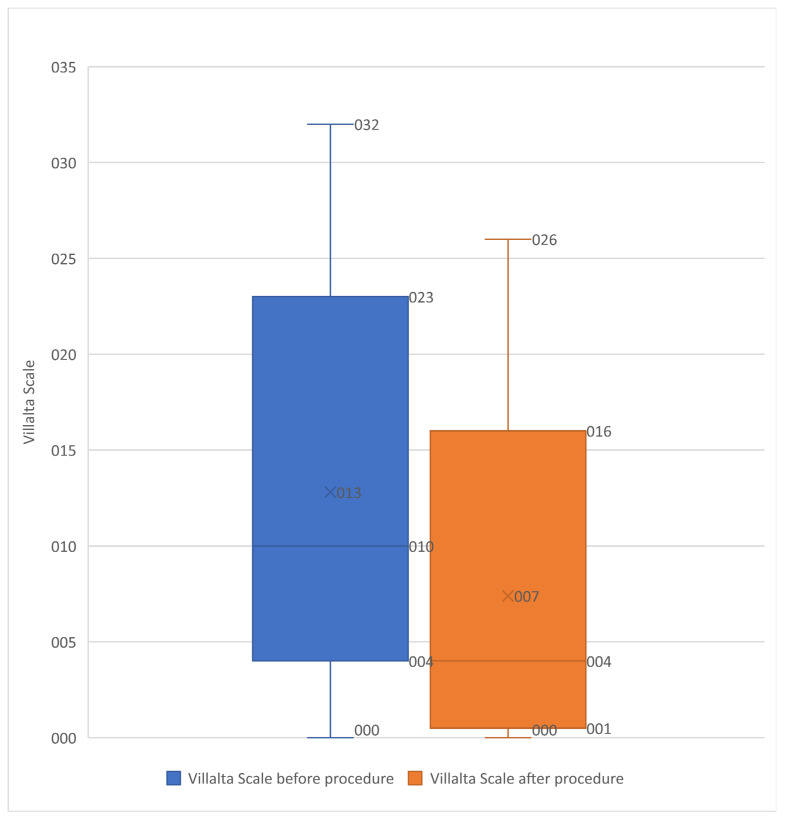
Box plot showing the Villalta scale before and after procedure in the 539 included patients. Box plot showing the Villalta scale improvement after procedure (significant improvement *p* < 0.0001).

**Figure 3 diagnostics-13-02357-f003:**
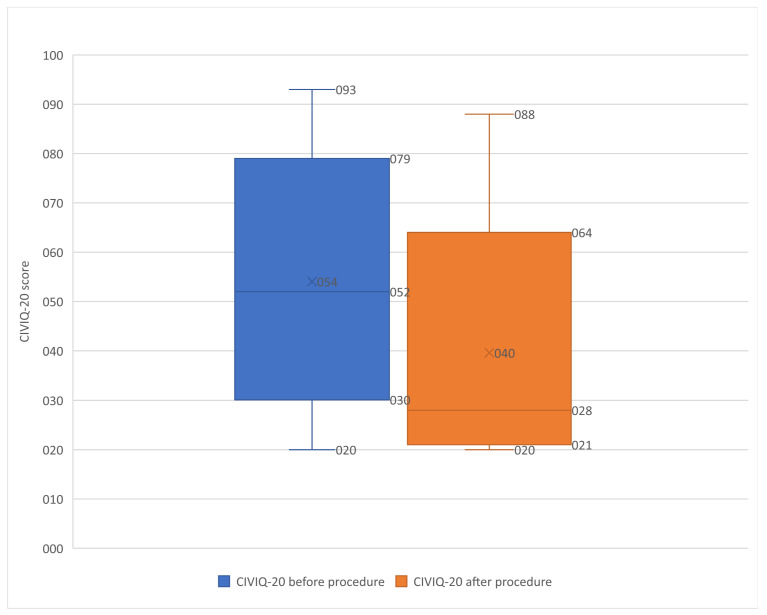
Box plot showing the CIVIQ-20 score before and after procedure in the 298 patients. Box plot showing the CIVIQ-20 score before and after procedure in the 539 included patients (significant improvement *p* < 0.0001). The remaining 241 patients were not screened for the CIVIQ-20 score during follow-up.

**Table 1 diagnostics-13-02357-t001:** Main features of the 539 patients for whom Villalta scores were available.

**Age (years), median [IQR]**	43 [32.3–56.5]
Females, *n* (%)	324 (60.1)
**Time from DVT to stenting (months), median [IQR]**	3.1 [1.3–11.9]
**DVT recurrence, *n* (%)**	
One	96 (17.8)
Two	30 (5.6)
Three or more	25 (4.6)
**DVT sequelae grade, *n* (%)**	
0	198 (36.7)
1	124 (23.0)
2	171 (31.7)
3	46 (8.5)
**Side of stenting, *n* (%)**	
Left	379 (70.3)
Right	78 (14.5)
Both	82 (15.2)
**Baseline Villalta score, median [IQR]**	12 [8–14]
**Baseline Villalta score category, *n* (%)**	
No PTS	31 (5.8)
Mild PTS	181 (33.6)
Moderate PTS	193 (35.8)
Severe PTS	134 (24.9)
**Baseline CIVIQ-20 score, median [IQR]**	52 [40–65]

IQR: interquartile range; PTS: post-thrombotic syndrome; DVT: deep vein thrombosis; Sequelae grade developed by Grenoble teams [18], CIVIQ-20: 20-item ChronIc Venous dIsease quality-of-life Questionnaire.

**Table 2 diagnostics-13-02357-t002:** Stenting in the 539 patients for whom Villalta scores were available.

**Stented Segments (LET Classification), *n* (%)**	
One	11 (2.0)
Two	21 (3.9)
Three	96 (17.8)
Four	215 (39.9)
Five	159 (29.5)
Six	36 (6.7)
Seven	1 (0.2)
**Distal stenting zone**	
Inferior vena cava	11 (2.0)
Common iliac vein	100 (18.6)
External iliac vein	366 (67.9)
Common femoral vein	52 (9.7)
**Immediate technical success, *n* (%)**	530 (98.3)
Post-procedural Villalta score, median [|IQR]	3 [1,2,3,4,5,6]
**Post-procedural Villalta score category, *n* (%)**	
No PTS	335 (62.2)
Mild PTS	139 (25.8)
Moderate PTS	40 (7.4)
Severe PTS	25 (4.6)
**Post-procedural CIVIQ-20 score, median [IQR]**	28 [22–40]

PTS: post-thrombotic syndrome; IQR: interquartile range; LET: lower extremity thrombosis classification; CIVIQ-20: 20-item ChronIc Venous dIsease quality-of-life Questionnaire.

**Table 3 diagnostics-13-02357-t003:** Factors independently associated with a Villalta score improvement after stenting.

	*p* > |t|	95% Confidence Interval	
**DVT sequelae grade**			
1	0.982	−1.025	1.048
2	0.244	−0.449	1.761
3	0.029	0.177	3.21
**DVT recurrence**			
One	0.397	−0.669	1.6851
Two	0.496	−1.295	2.670
Three	0.487	−1.929	4.047
Four	0.078	−0.307	5.794
Five	0.764	−3.394	2.493
Six or seven	0.000	−11.248	−4.527
**Stented segments, *n***			
One	0.164	−6.201	1.052
Two	0.377	−4.922	1.866
Three	0.356	−4.802	1.728
Four	0.653	−4.093	2.568
Five	0.241	−5.738	1.446
Six or seven recurrences	0.007	1.290	8.316
**Sex**	0.135	−1.486	0.201
**Age**	0.18	−0.047	0.009
**Median time from DVT to stenting**	0.001	−0.108	−0.027

**Table 4 diagnostics-13-02357-t004:** Factors independently associated with a CIVIQ-20 score improvement after stenting.

	*p* > |t|	95% Confidence Interval	
**Thrombosis recurrence, *n***			
One	0.001	−11.107	−2.874
Two	0.924	−7.2136	7.947
Three	0.367	−15.726	5.835
Four	0.536	−14.716	28.224
Five	0.649	−4.887	3.050
Six or seven	0.891	−9.382	10.793
**Stented segments, *n***			
One	0.042	0.743	42.574
Two	0.229	−7.208	30.044
Three	0.180	−6.107	32.438
Four	0.332	−9.837	29.053
Five	0.219	−7.751	33.669
Six	0.513	−13.911	27.790
**Stenting distality ^a^**			
Inferior vena cava	0.173	−29.269	5.279
Common iliac vein	0.516	−21.529	10.842
External iliac vein	0.431	−23.652	10.112
**Age**	0.189	−0.183	0.037
**Bilateral thrombosis**	0.002	4.0137	17.475

^a^ The common femoral vein was omitted given the small number of cases.

## Data Availability

Not applicable.

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
