# Peer review of "Patient-Reported Outcomes of Endovascular Treatment of Post-Thrombotic Syndrome: Ancillary Study of a French Cohort"

_diagnostics, 2023, doi:10.3390/diagnostics13142357_

Round 1

Reviewer 1 Report

The article under review presents a detailed retrospective examination of outcomes following angioplasty/stenting for Post-Thrombotic Syndrome (PTS) in a French cohort. The study's primary strength is its comprehensive approach to data analysis, focusing on both objective and patient-reported outcomes through the Villalta scale and the Chronic Venous Disease quality-of-life Questionnaire (CIVIQ-20).

The authors provide findings from a sample size of 539 patients, followed over a decade (2009–2019).

The investigators reveal a correlation between the degree of Villalta-scale improvement and variables such as the severity of thrombosis sequelae and time from thrombosis to stenting. Simultaneously, they illuminate the connection between CIVIQ-20 score improvement and factors like bilateral stenting, single thrombosis recurrence, and single stented segment.

This dual approach to measuring improvement - objective criteria and patient-reported outcomes - stands out as a novel and insightful method, highlighting the divergence between the two types of outcomes and identifying different factors associated with each. While the objective gains were already reported in previous works, this study goes further in illustrating patient-reported improvements.

A potential area of concern is that the CIVIQ-20 score was only available for 298 out of the 539 patients, which could introduce some level of bias or inaccuracies in the overall patient-reported results. However, the authors appear to address this adequately through rigorous statistical analysis.

The presentation and interpretation of the results are clear and straightforward, with the authors effectively explaining complex concepts and analyses.

Line 294 “Because of the heterogeneity of the treatment strategy and linked with pass medical history only systematic review are published.” Should read like Due to the heterogeneity of the treatment strategy and its connection to past medical history, only systematic reviews are published.

Line 308 The ambiguous statement 'Due to the heterogeneity of the treatment strategy and its relationship with past medical history, only systematic reviews are published' needs further clarification. Additionally, the statement on line 310 requires grammatical and contextual corrections. Indeed most data with regard to combined anticoagulation and antiplatelet treatments originate from studies of patients with coronary heart disease (PMID: 36568540) but as both bleeding and acute or chronic reocclusion may influence patient reported outcomes this topic requires more detailed discussion.

Reviewer 2 Report

Review Report for the Manuscript “Patient-Reported Outcomes of Endovascular Treatment of Post- Thrombotic Syndrome: Ancillary Study of a French Cohort

Rating the Manuscript

Originality/Novelty: Is the question original and well defined? Do the results provide an advance in current knowledge?

Yes, in the manuscript the authors have reviewed the Villalta scale and 20-item ChronIc Venous dIsease quality-of-life Questionnaire (CIVIQ-20) scores recorded after endovascular stenting for for post-thrombotic syndrome.

Significance: Are the results interpreted appropriately? Are they significant? Are all conclusions justified and supported by the results? Are hypotheses and speculations carefully identified as such?

Yes, the results are interpreted well.

Quality of Presentation: Is the article written in an appropriate way? Are the data and analyses presented appropriately? Are the highest standards for presentation of the results used?

Yes, the article is written well. The representation of data could be improved. Quality of the figures, figure captions could be improved.

Scientific Soundness: is the study correctly designed and technically sound? Are the analyses performed with the highest technical standards? Are the data robust enough to draw the conclusions? Are the methods, tools, software, and reagents described with sufficient details to allow another researcher to reproduce the results?

Yes, the data is robust enough to draw conclusions and the methods, tools and methods used in the data analysis are explained properly.

Overall Merit: Is there an overall benefit to publishing this work? Does the work provide an advance towards the current knowledge? Do the authors have addressed an important longstanding question with smart experiments?

Yes. This study provides an advancement to the current knowledge. 

English Level: Is the English language appropriate and understandable?

Yes, English language in the manuscript is appropriate and understandable. 

Overall Recommendation: This is a well written paper and I only have few comments. 

Accept after Minor Revisions

Given below are the comments for each section of the manuscript.

Abstract

The abstract is written and summarizes the content of the manuscript.

Introduction

Line 96: “Abnormalities of the microcirculation and lymphatic system also develop.”

Briefly mention what type of abnormalities could develop.

Line 93: “They include congestive signs. A specific quality-of-life assessment tool for chronic venous insufficiency, the 20-item ChronIc Venous dIsease quality-of-life Questionnaire (CIVIQ-20), was developed in France and validated internationally”.

Are these the only two questionnaire/tools available for this? If there are any other methods, why did you select these two methods?

Materials and Methods:

2.2. Study population

Line 115: “Exclusion criteria were acute thrombosis, extrinsic compression by a tumour, non-thrombotic obstruction, Budd-Chiari syndrome, and venous occlusion at the site of a dialysis catheter.”

What are the reasons for selecting this exclusion criteria?

Results

Line 189: “All patients with CIVIQ-20 score data also had Villalta score data.”

It’s better if the authors could briefly mention about the advantages and disadvantages of these two criteria. 

Figures 

Figure 1: Caption for Figure 1 needs to be more informative.

Figure 2 and 3: For figures 2 and 3, could you include all the data points in the boxplots. I think if the authors present all the data points it would be more informative.

Also, I think the information in the chart title should go into the figure caption.

And authors need to label x and y axis. Include a legend describing the color codes used in the charts.

Discussion:

What would you do differently to overcome the limitations discussed in the last paragraph of the discussion section.

References:

Some of the references are more than 10 years old. It they don’t contain important information authors could replace these with new references. 

References: 5,6,7,8,10,16,17,26,27,28,37,40
